# Polarity-dependent nonlinear optics of nanowires under electric field

Regev Ben-Zvi [1,3], Omri Bar-Elli [2,3], Dan Oron [2✉] & Ernesto Joselevich [1✉]

Polar materials display a series of interesting and widely exploited properties owing to the inherent coupling between their fixed electric dipole and any action that involves a change in their charge distribution. Among these properties are piezoelectricity, ferroelectricity, pyro-electricity, and the bulk photovoltaic effect. Here we report the observation of a related property in this series, where an external electric field applied parallel or anti-parallel to the polar axis of a crystal leads to an increase or decrease in its second-order nonlinear optical response, respectively. This property of electric-field-modulated second-harmonic generation (EFM-SHG) is observed here in nanowires of the polar crystal ZnO, and is exploited as an analytical tool to directly determine by optical means the absolute direction of their polarity, which in turn provides important information about their epitaxy and growth mechanism. EFM-SHG may be observed in any type of polar nanostructures and used to map the absolute polarity of materials at the nanoscale.

---

[1] Departments of Materials and Interfaces, Weizmann Institute of Science, Rehovot, Israel. [2] Departments of Physics of Complex Systems, Weizmann Institute of Science, Rehovot, Israel. [3] These authors contributed equally: Regev Ben-Zvi, Omri Bar-Elli. ✉email: dan.oron@weizmann.ac.il; ernesto.joselevich@weizmann.ac.il

Polarity in compound semiconductors has been the focus of extensive study owing to its effect on both physical and surface properties[1,2]. Some of the most studied and utilized semiconductors, such as GaN, ZnO and CdSe possess a hexagonal Wurtzite (WZ) structure with a fixed dipole along the [0001] direction, also referred to as the polar c-axis. Crystals with polar orientation can grow along either the [0001] or [000$\bar{1}$] directions, then termed metal-polar or anion-polar, respectively, which have a strong influence on both their chemical and optoelectronic properties[1]. This effect is especially significant in 2D and 1D nanostructures due to their large surface-to-volume ratio. Specifically, polarity was found to influence chemical reactivity[3,4], doping and impurity incorporation[5–7], surface states and band-bending, which can strongly affect the nature of metallic contacts in functional systems, reflectivity[8] and the performance of devices based on polar orientations, such as LEDs and high-mobility transistors[1] (HEMTs). In nanowires, the polarity is a key property[9], as it was found to affect morphology[10,11], growth rates[12] and defect incorporation; and therefore has a significant impact on their electronic structure and optical properties[13]. Determination of the polarity of nanowires has been achieved by advanced transmission electron microscopy (TEM) methods, such as high-angle annular dark field (HAADF) and annular bright field (ABF) imaging along the nanowires[14,15].

During the last decade, there has been a growing interest in the production of planar nanowire arrays by exploiting the epitaxial relations between the nanowires and a crystalline substrate, by surface-guided catalytic vapour–liquid–solid (VLS) growth[15–23] and recently also by selective area epitaxy[24–26]. Surface-guided growth has been shown to enable the production of planar nanowires with a large variety of controlled crystallographic orientations, including zincblende (ZB)- and wurtzite (WZ)-structure semiconductor nanowires oriented along polar, non-polar and semipolar directions[17,21,22]. These surface-guided approaches eliminate the need for post-growth assembly, and were found to be general to many materials and to produce high quality, single-crystal nanowires[17–22,27], readily arranged for integration into various functional systems, including logic circuits[28], photodetectors[17,20,27,29], photovoltaic cells[30] and prospective quantum computing circuits[24–26].

Determining the crystallographic orientation of these planar nanowires by TEM is especially challenging due to their strong (usually covalent) attachment to the substrate, which requires production of cross-sectional lamellae of selected nanowires using focused-ion beam (FIB) milling[22]. Recently, we have shown that the crystallographic orientation of surface-guided nanowires can be efficiently and non-invasively mapped by an optical method based on SHG polarimetry[31]. For instance, using this method, we showed that surface-guided ZnO nanowires with a polar orientation can be obtained with high selectivity by growing them on R (1$\bar{1}$02) plane sapphire (α-Al$_2$O$_3$), whereas on M (10$\bar{1}$0) plane sapphire the ZnO nanowires grow in either a polar or nonpolar orientation depending on their growth direction with respect to the substrate crystal lattice. Using SHG polarimetry, we can clearly distinguish between polar, nonpolar or semipolar nanowires[31–33]. However, determining the direction (i.e. the absolute sign) of the polarity of polar nanowires has proved challenging. As mentioned above, the direction of polarity can be obtained from high-resolution TEM analysis of a lamella cut along the wire. Yet, such lamellae are difficult to obtain and to image, making this process very low throughput, or impractical, for certain samples.

A simple and robust optical method to determine nanowire polarity would thus be highly desirable. Unfortunately, only the SHG phase (and not its intensity) is sensitive to the direction of polarity of the sample, because the electric field of light is symmetrical in opposite directions within the plane of polarization. A phase-sensitive measurement is thus required to distinguish between nanowires that have, say, [0001] and [000$\bar{1}$] polar directions. One potential way to measure this phase is via interferometric second-harmonic imaging, whereby a second-harmonic field from a reference crystal is interfered with SHG either at the detector or at the sample[34–37]. However, this type of measurement typically does not provide an absolute characterization of the direction of polarity but rather a relative phase measurement across the sample, and requires maintaining interferometric stability of the entire optical setup, which poses significant technical challenges (these can be partially alleviated using inline interferometry[38]). Determination of the absolute direction of polarity in ferroelectrics has been performed by electro-optic interferometric imaging, whereby a modulated external electric field (providing the absolute direction) is used to phase modulate a laser beam transmitted through the sample via the well-known Pockels effect[39]. This phase modulation is, similarly to phase-sensitive SHG, detected via an interferometric measurement. The requirement for interferometric measurements has dramatically hindered the use of either of these techniques, particularly as they suffer from additional drawbacks relating to the difficulty to apply interferometric measurements in a backscattering geometry, and the sensitivity of SHG measurements to dispersion and chromatic aberrations. Here we devise an alternative phase-sensitive measurement that does not rely on the use of an external reference but rather on inducing interference between two different nonlinear processes within the sample using an external electric field. This eliminates the stability requirements while providing an absolute spatial direction of the measured polarity. To do this, we apply a slowly varying external electric field along the nanowire axis, which induces an increase or decrease of the SHG signal depending on the direction of the applied electric field with respect to the nanowire, enabling determination of its polarity.

Indeed, it is well established that SHG can be induced in a centrosymmetric crystal by applying an external electric field that breaks its inversion symmetry. This method, known as electric-field-induced second harmonic (EFISH), was first demonstrated for calcite[40]. The same principle can be applied to non-centrosymmetric crystals such as CdS[41] and GaAs[42] by applying an electric field along the nonpolar [1$\bar{2}$10] direction of the WZ structure or the [001] direction of the ZB structure, where SHG is symmetry forbidden. However, to the best of our knowledge, the modulation of SHG along the polar axis of nanostructures by an external electric field due to interference between the static SHG and EFISH component has not yet been reported, and the ability to exploit this phenomenon as a tool to determine polarity has not yet been demonstrated. We note that electric field modulation of SHG in centrosymmetric materials exhibiting SHG of an electric quadrupole or magnetic dipole origin does not lead to such an interference since in this case, the static and EFISH components are at quadrature with each other for non-absorbing media[43].

In this work, we demonstrate the modulation of SHG by applying an external electric field along the polar axis of a single nanowire, and show that the sign of this modulation depends on the direction of the nanowire polarity with respect to the field. The measurement is done in a lock-in technique by applying a low-frequency AC electric field, to enable the detection of a small modulation on top of the relatively intense SHG signal. The sign of polarity is determined by the phase between the SH optical signal and the applied electric field AC modulation. Moreover, this method provides a quantitative assessment of the relative magnitude of the second- and third-order nonlinear susceptibilities, which could be used for mapping the homogeneity of the

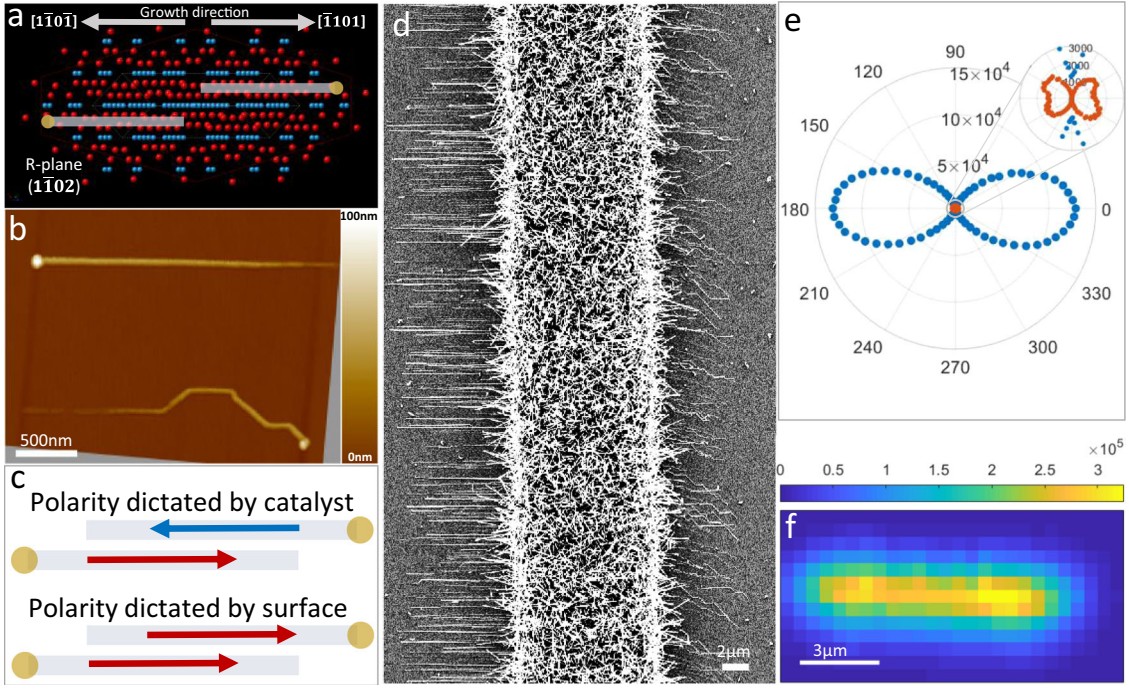

**Fig. 1 Surface-guided ZnO nanowires on R-plane sapphire: epitaxy, SHG mapping and polarimetry. a** Crystallographic model of R ($1\bar{1}02$) plane sapphire showing clear atomic rows of Al and O along ±[$1\bar{1}0\bar{1}$] that give rise to the nanowires' alignment in these directions. The two cartoons depict the two possible growth directions of nanowires. **b** AFM image of two nanowires growing in two opposite directions on R-plane sapphire from gold nanoparticles, presenting the different morphologies between the two possible growth directions (straight to the left and kinked to the right). **c** Scheme describing the two possible outcomes. 1. Catalyst droplet dictates polarity so that nanowires with opposite growth directions have the same polarity and 2. Substrate dictates polarity so that nanowires with opposite growth directions have opposite polarities (the red and blue arrows denote polarity to the right and left, respectively, and the yellow circles represent the catalyst droplet (the growing tip of the nanowire). **d** SEM image of ZnO nanowires growing in two opposite directions on R-plane sapphire from evaporated gold catalyst. In one direction the nanowires are perfectly aligned while in the other they tend to lose directionality. **e** Typical SHG polarimetry of a polar nanowire along either the [$1\bar{1}0\bar{1}$] or the [$\bar{1}101$] direction of the sapphire, depicting the excitation polarization dependence of the SHG signal. The radial coordinate is the signal intensity (in counts per second) and the angle is the excitation polarization angle in the lab frame. Inset is a magnification of the centre of the plot. **f** SHG mapping at a fixed polarization (signal colour scale in arbitrary units). Note that all plates (**a–f**) are shown in the same orientation with respect to the [$1\bar{1}0\bar{1}$] and [$\bar{1}101$] directions of the substrate.

structure. Applying this non-invasive method, we determine the absolute sign of polarity of ZnO nanowires lying on a substrate. This information allows us to answer an important lasting question regarding surface-guided nanowires: whether their polarity is dictated by the symmetry of the underlying substrate or by their growth direction. This question is fundamentally important regarding the guided growth mechanism and the interaction of the catalyst with the substrate and the growing nanowires, and is practically important regarding our ability to produce arrays of nanowires with coherent polarities for optoelectronic applications. A quantitative study of how the nonlinear electro-optical phenomenon of EFM-SHG scales with the intensity of the applied electric field, in combination with a simple theoretical model, allows us to understand the physical nature of the phenomenon. EFM-SHG is shown to serve as a crystallographic characterization tool to determine nanowire polarity. In principle, this phenomenon is not limited to surface-guided nanowires or any nanowire or nanostructure, but could be used as a general, fast, simple and non-destructive method for polarity mapping and quantitative assessment of the relevant second-order nonlinear susceptibility of any material at the nanoscale, with extensive applications in materials science and nanotechnology.

## Results

### Crystallographic orientation and polar effects in guided nanowires.

Surface-guided ZnO nanowires were grown by chemical vapour deposition (CVD) on R ($1\bar{1}02$) plane sapphire in a VLS mechanism from Au catalyst (see 'Methods'). Surface-guided ZnO nanowires tend to grow with preferable alignment along the ±[$1\bar{1}0\bar{1}$] directions of the R-plane sapphire as depicted in the atomic model in Fig. 1a. Horizontal alignment on R-plane sapphire, with an unambiguous preferred alignment along ±[$1\bar{1}0\bar{1}$], is common to many systems such as the growth of carbon nanotubes[44,45], etching of graphene by metallic droplets[46] and the growth of surface-guided nanowires[22]. This preferred direction for alignment arises from the highly ordered arrangement of atoms on the surface of R-plane sapphire (Fig. 1a) with rows of Al and O atoms along the ±[$1\bar{1}0\bar{1}$] direction, which lead to an anisotropic interaction of the substrate and the metal catalyst, making the movement along this direction energetically favourable. Scanning electron microscopy (SEM) images (Fig. 1d) show surface-guided ZnO nanowires on R-plane sapphire, growing from a microscale stripe of evaporated gold. The nanowires grow along the common two opposite directions (±[$1\bar{1}0\bar{1}$]) but, while in one direction they are perfectly aligned, in the other one they tend to grow in a kinked morphology with short segments deviating at nearly ±40° from the [$\bar{1}101$] direction.

This observed asymmetry in the nanowire alignment on R-plane sapphire is explained by the dissimilarity of the [$1\bar{1}0\bar{1}$] and the [$\bar{1}101$] directions, which for other materials results in unidirectional growth, as in the case of CNTs[47] and guided GaN nanowires[22]. Figure 1b presents an AFM image of two ZnO nanowires growing on R-plane sapphire from Au nanoparticles in a similar manner, further demonstrating the asymmetric growth

in the non-equivalent opposite directions $[1\bar{1}0\bar{1}]$ and $[\bar{1}101]$. The growth direction of the guided nanowires is determined according to the VLS mechanism, in which the catalyst droplet is pushed further away while the nanowire crystalizes. The location of the catalyst droplets at the edges indicate two opposite growth directions, as clearly seen in the AFM image.

In general, the polarity of nanowires is directly affected by the growth conditions and the growth mechanism. For example, it was found that unlike self-catalysed ZnO nanowires, which mostly grow along their Zn-polar orientation [0001], Au-catalysed ZnO nanowires often grow in their O-polar orientation $[000\bar{1}]$ due to the important role of the Au nanoparticle in the VLS growth mechanism[48]. The anion-polar growth direction is common to many other nanowire materials such as GaN, GaAs, ZnTe etc. In surface-guided VLS nanowires, the epitaxial relations with the underlying substrate add another level of complexity and can play an important role in polarity determination. As shown above, R-plane sapphire supports two opposite growth directions, which differ in epitaxy. Thus, in principle, the nanowires' polarity may be dictated either by the growth direction (catalyst) or by the substrate's epitaxial guidance. The former would result in all nanowires having the same polarity orientation with respect to growth direction, and thus opposite direction on the substrate (Fig. 1c top), while the latter would result in nanowires of opposite growth direction to have opposite polarity with respect to growth direction (Fig. 1c bottom). To answer this important scientific question, we need to determine the polarity of many ZnO nanowires efficiently enough to be able to check whether they correlate with the crystallographic orientation of the sapphire or with their growth direction. An optical method to determine nanowire polarity would be ideal for this purpose, while the conventional TEM-based methods would be very laborious, if not practically impossible.

**Optical determination of nanowire polarity.** In previous works, we showed that surface-guided ZnO nanowires on R-plane sapphire grow along the ZnO crystal's c-axis, corresponding to the polar ±[0001] axis of the WZ structure[21,31]. To map the crystallographic orientations of the nanowires, we employed SHG polarimetry, where a rotation of the excitation polarization in the plane of incidence results in a change of the measured SHG intensity (see 'Methods'). In short, the induced second-harmonic polarization in the crystal may be described by[49]:

$$P(2\omega) = 2\varepsilon_0 d \cdot \mathbf{E}^2(\omega) \tag{1}$$

Where $\mathbf{E}$ is the laser field ($\mathbf{E}^2 = E_x^2, E_y^2, E_z^2, 2E_yE_z, 2E_xE_z, 2E_xE_y$), and $d$ replaces the second-order susceptibility tensor $\chi^{(2)}$. In the case of the WZ structure and under Kleinman symmetry there are only two independent elements in this "d-matrix":

$$d_{WZ} = \begin{pmatrix} 0 & 0 & 0 & 0 & d_{15} & 0 \\ 0 & 0 & 0 & d_{15} & 0 & 0 \\ d_{15} & d_{15} & d_{33} & 0 & 0 & 0 \end{pmatrix} \tag{2}$$

The non-zero elements and their ratio in the d-matrix define the patterns seen in the polar plots in Fig. 1e. In particular, the alignment of the c-axis of the crystal with respect to the growth axis may be extracted. For polar nanowires studied here, the c-axis of the crystal coincide with the nanowire's long axis and lies on the x-axis in the lab frame which corresponds with the 0° in the polar plots. Imaging (mapping) is done by measuring the SHG intensity at a fixed polarization while scanning the position of a piezo stage (Fig. 1f) (see 'Methods'). Our statistical polarimetry study showed that all surface-guided ZnO nanowires on R-plane sapphire grow along their polar axis, regardless of

their growth direction. However, the absolute sign of polarity, that is, whether the surface-guided nanowires are growing along the ZnO [0001] or $[000\bar{1}]$ direction, cannot be resolved via polarimetry. This is due to the $C_2$ rotation symmetry of light polarization, namely that 0° and 180° polarizations are identical.

In general, the second-order response of a non-centrosymmetric polar medium is determined by the second-order nonlinear susceptibility tensor $\chi^{(2)}$ (as represented by the d-matrix of Eq. (2)). The elements of this tensor are determined by the crystal symmetry, electronic structure and the wavelength. The general derivation of the zzz component (relevant for this work) from fundamental principles is complicated, but $\chi^{(2)}_{zzz}$ values are tabulated for numerous materials, including ZnO. An external electric field applied on the medium modifies its nonlinear response as it can further modify the symmetry of the medium. Since here the field is applied along the polar axis, this process is characterized by the third-order nonlinear susceptibility tensor element $\chi^{(3)}_{zzzz}$, which is closely related to the nonlinear refractive index (Kerr coefficient). Notably, for most materials under nonresonant excitation (that is, for wavelengths well below the band gap), including ZnO, $\chi^{(3)}_{zzzz}$ is positive[50]. Mathematically, the nonlinear polarization at the second harmonic under an applied field is described by the interference of these two terms:

$$P(2\omega) = \chi^{(2)}(2\omega, \omega, \omega)E^2(\omega) + \chi^{(3)}(2\omega, \omega, \omega, 0)E^2(\omega)E_{AC}(\omega_{AC}) \tag{3}$$

Where $\chi^{(2)}$ is the second-order susceptibility, $E(\omega)$ is the high-frequency electric field of the laser, $\chi^{(3)}$ is the third-order susceptibility and, $E_{AC}(\omega_{AC})$ is the low-frequency applied AC electric field. The first term is the standard second-order response responsible for the SHG, while the second term is the well-known third-order EFISH term, which must be included due to the presence of an additional, external, electric field. The situation described by Eq. (3) resembles that reported for oriented molecular layers when the molecular dipole is modified by a change of the pH difference across the layer[51,52].

Taking the square of the induced polarization, the detected signal at the SH frequency is then given by:

$$I(2\omega) \propto |P(2\omega)|^2 = |E^2(\omega)|^2 \left\{ [\chi^{(2)}]^2 + 2\chi^{(2)}\chi^{(3)}E_{AC}(\omega_{AC}) + [\chi^{(3)}]^2 |E_{AC}(\omega_{AC})|^2 \right\} \tag{4}$$

Where we dropped the explicit frequency dependence of $\chi^{(2)}$ and $\chi^{(3)}$ for clarity. Here we have three terms contributing to the measured SHG under an external electric field. The first does not oscillate with the external AC field. The second, which is due to the interference of the SHG field and the EFISH field, oscillates with the same frequency—$\omega_{AC}$, and gives rise to EFM-SHG. The third has two components, a constant offset to the non-oscillatory term, and one which oscillates with a doubled frequency—$2\omega_{AC}$ (since: $\sin^2\alpha \propto \cos^2\alpha$). Importantly, the sign of the AC field only affects the sign of the second term, giving rise to an in-phase or an anti-phase oscillation, depending on the relative sign of the second and third-order nonlinear susceptibility terms. For ZnO, studied here, the relative sign is such that an increase of the SHG signal is expected when the electric field points along the internal dipole of the ZnO nanowire[53–55].

For nonlinear electro-optical measurements, surface-guided ZnO nanowires were grown on R-plane sapphire from Au nanoparticles, deposited from a dilute dispersion, to form an even coverage of nanowires growing in both direction on the sample. In order to apply an electric field parallel to the polar axis of the ZnO crystal, an array of gold electrodes with a 10-μm gap was patterned (see 'Methods') according to the growth direction of the nanowires. The nanowires' lengths ranged from ~3 to 8 μm and

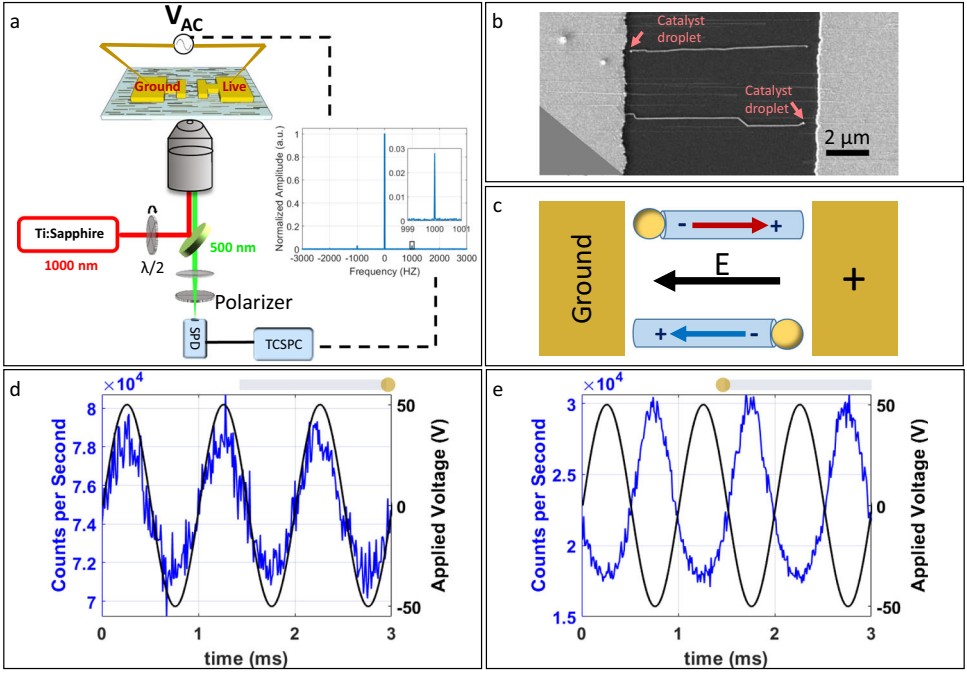

**Fig. 2 Electric-field-modulated SHG of nanowires with opposite growth directions. a** Measurement scheme. 1000-nm laser is coupled into an inverted microscope and focused on a single nanowire lying between two electrodes. The SHG signal at 500 nm is collected by a single-photon detector (SPD) and time stamped using a time-correlated single-photon counting (TCSPC) system. A low-frequency (1 kHz) sine wave of various voltage amplitudes is applied. FFT spectrum of the SHG time trace collected while supplying the 1 kHz AC voltage. Inset: zoom in on the amplitude at 1 kHz. **b** SEM image of two nanowires with opposite growth directions between a pair of patterned electrodes. **c** A scheme of the configuration shown in (**b**). The red and blue arrows indicate the polarity of nanowires in opposite directions. **d** Applied AC voltage (black) and the time-averaged optical SHG signal (blue) from a right-growing nanowire and (**e**) a left-growing nanowire showing in-phase and anti-phase oscillations.

the spacing between adjacent nanowires was typically a few microns (see 'Methods'), enabling a contactless configuration and optical probing of a single nanowire at a time (Fig. 2b). Mapping with SEM enabled the imaging of electrode pairs containing nanowires in the gap and determining the growth direction of each nanowire by locating the catalyst droplet at its edge. Samples were placed face up on a cover slip to allow needle probe access for applying voltages (Fig. 2a). Before applying the electric field, SH maps and polarimetry data from several nanowires were acquired as described above (Fig. 1e, f). Mapping ensured access to a specific point, approximately at the centre of the nanowire and far enough from the electrodes to exclude scattering from the surfaces of the electrodes themselves. To study the modulation of SHG in response to an applied electric field, the SH intensity from a single nanowire was measured while an AC voltage was applied to the relevant electrodes (Fig. 2a). This measurement was done using a fixed laser polarization, where both excitation and collection are parallel to the growth axis of the nanowire (and to its $c$-axis), enabling access to $\chi_{zzz}^{(2)}$ and $\chi_{zzzz}^{(3)}$. Typically, a 1 kHz sine wave of various voltage amplitudes was supplied using an analogue voltage output device (see 'Methods').

Figure 2d, e shows the SH optical signal (blue) (averaged in time, see 'Methods') with respect to the supplied AC voltage (black) for two nanowires with opposite growth directions. While the SH signal of the nanowire growing to the right shows an in-phase response (Fig. 2d), the signal from the left-growing nanowire shows an anti-phase response (Fig. 2e). The electric field is applied in the same manner in both measurements, so that the measured polarity of the two nanowires is opposite with respect to the applied field. Since the nanowires are growing in opposite directions, we conclude that they exhibit the same polarity with respect to the growth direction (Fig. 1c, top). However, obtaining such time-averaged signals is time consuming, requiring several minutes of

integration. A more rigorous approach was implemented by using FFT analysis (see 'Methods') yielding similar results to a standard lock-in detection scheme. In Fig. 2a an example FFT spectrum is given where a clear optical response is observed at the same frequency as the applied AC field (1 kHz in this case), indicating an electric-field-modulated SHG. Extracting the phase of the oscillation at 1 kHz from the FFT spectrum requires only a few seconds of integration. Furthermore, the FFT analysis facilitates detection of oscillations at less pronounced frequencies that, according to Eq. (2), should be present.

**Polarity of surface-guided nanowires**. We performed the same phase analysis on eighteen different nanowires. Results are presented in Table S1. SEM imaging with a secondary-electron detector was used to determine the growth direction of the inspected nanowires by identifying the catalyst droplet at their end (Figs. S2–3). Out of eighteen nanowires, nine were found to grow to the right, and showed an in-phase response, and eight were found to grow to the left, and showed anti-phase response. Only in one of the eighteen nanowires, the measured phase did not conform to the growth direction. This may be due to the difficulty of precisely spotting the catalyst nanoparticle at the end of each nanowire in a few cases (Figs. S2–3). This important observation demonstrates that the ZnO nanowires all grow with the same polarity with respect to their growth direction, as depicted in option I (top) of Fig. 1c, whereas the asymmetry of the substrate is found to not affect the polarity of the surface-guided nanowires.

After establishing that the polarity of surface-guided ZnO nanowires is dictated by their growth direction and not by the lack of a symmetry (mirror or glide) plane across the growth direction in the R-plane sapphire, we can turn to determine their absolute polarity. As discussed above, an in-phase response

means for ZnO that positive values of voltage result in an increase of the SH intensity, while negative values of the voltage result in a decrease of the SH intensity. This means that positive voltage leads to a parallel configuration between the electric field and the internal dipole of the ZnO crystal. Similarly, negative voltage leads to an anti-parallel configuration between the electric field and the internal dipole of the ZnO crystal. The same logic can be applied to the anti-phase response, observed for the nanowires growing to the left (i.e. $[1\bar{1}0\bar{1}]$). This observation indicates a crystal dipole $[0001]$ pointing away from the catalyst droplet in both cases (Fig. 2c). In other words, this observation indicates growth towards the $[000\bar{1}]$ direction, meaning that surface-guided ZnO nanowires are oxygen-polar. Because the polar ZnO nanowires can grow in opposite directions $\pm[1\bar{1}0\bar{1}]$, and their polarity is dictated by the growth direction rather than by the asymmetry of the substrate, both epitaxial relations $[000\bar{1}]_{ZnO}||$ $[1\bar{1}0\bar{1}]_{Al2O3}$ and $[000\bar{1}]_{ZnO} || [\bar{1}101]_{Al2O3}$ coexist in the ensemble, but the former is relatively stronger than the latter, as indicated by their respective straight vs. kinked morphologies. The kinks that appear only in the nanowires that grow along the $[\bar{1}101]$ direction of the sapphire, usually appear as close pairs encompassing short nanowire segments whose directions deviate by nearly $\pm40°$ from the $[\bar{1}101]$ direction of the longer segments. This indicates that although the preferred growth direction is along $[\bar{1}101]_{Al2O3}$, this preference is not as strong as along the opposite $[1\bar{1}01]_{Al2O3}$ direction. The nearly $\pm40°$ deviation from $[\bar{1}101]_{Al2O3}$ is consistent with the $\{20\bar{2}1\}_{Al2O3}$ directions that were found to be preferred by other Wurtzite-structure nanowires like ZnS[19], ZnSe[17] and CdSe[20]. This means that although the catalyst determines the polarity of the nanowires, the interaction with the surface does still have a significant effect, and there could be a competition between surface-determined and growth direction-

determined polarity. Although the latter wins for ZnO NWs on R-plane sapphire, it might be different for other material-substrate combinations. The kinked morphology could in principle affect the optical signal. However, since the kinked segments are shorter than the ones along the main direction, we opted to neglect this possible effect, which could greatly complicate the analysis.

The polarity of vertical ZnO nanowires with respect to their growth direction was reported to depend on the growth mechanism, usually being Zn-polar for catalyst-free nanowire growth and O-polar for VLS nanowire growth. The important role of the catalyst droplet in determining nanowire polarity, and the mechanism that leads to it as opposed to catalyst-free growth have been recently reviewed in detail[9]. Moreover, other VLS-grown nanowires of compound semiconductors[14] tend to grow anion-polar like ZnO. The tendency of vertical VLS nanowires to grow as anion-polar in a variety of materials suggests a general phenomenon, related to the catalyst droplet and not to the underlying substrate. When it comes to surface-guide nanowires, the polarity seems to be dictated by the growth direction similarly as in vertically grown VLS nanowires. On the other hand, the substrate still plays an important role in determining the growth direction and the crystallographic orientation of surface-guide nanowires, as described before[17,21,22].

**Quantitative analysis of electric-field-modulated SHG.** We now try to better understand the EFM-SHG phenomenon by quantitatively analysing its scaling with the intensity of the applied electric field. The time trace of the SH signal (Fig. 2d, e) clearly shows that the optical signal oscillates at the same frequency as the applied electric field. However, the FFT spectrum of the collected photons (Fig. 3a) clearly shows an additional term at twice that

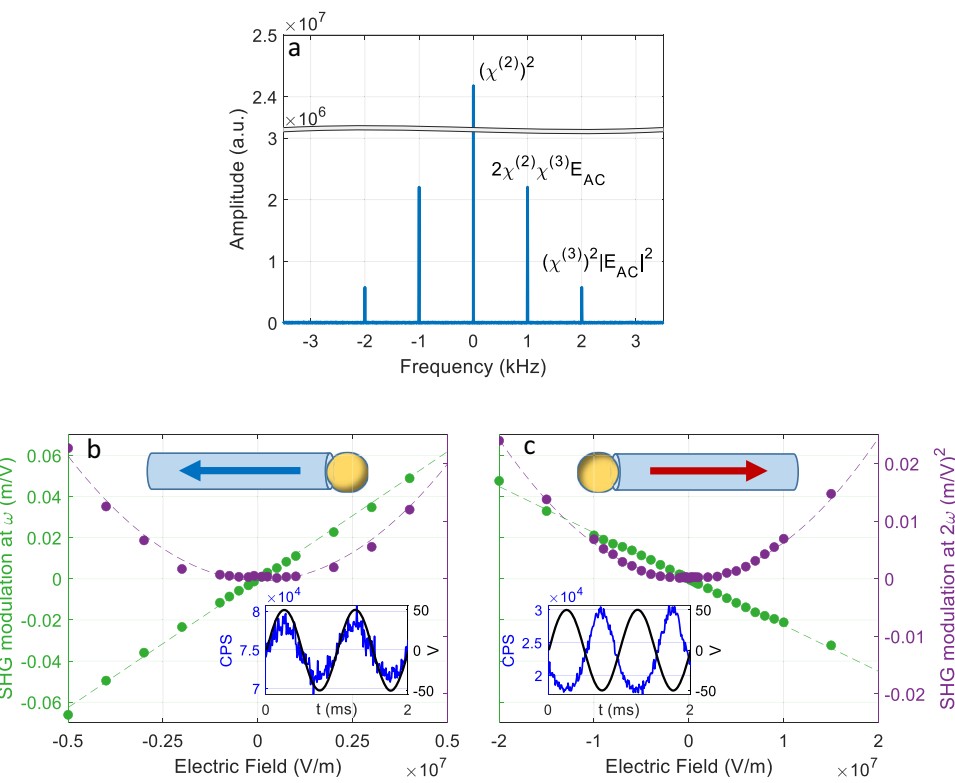

**Fig. 3 Voltage-dependent EFM-SHG measurements. a** FFT spectrum of an EFM-SHG measurement. The three frequency components are labelled according to Eq. (4). **b** The amplitudes at 1 kHz (green) and 2 kHz (purple) as a function of external applied filed strength measured from a right-growing nanowire. The solid lines are linear and quadratic trend lines. **c** The same as in (**b**) but for a left-growing nanowire. The insets in (**b**) and (**c**) show the time-averaged SHG signal (blue) in counts per second (CPS) with respect to the applied AC voltage (black), as in Fig. 2e, d, respectively.

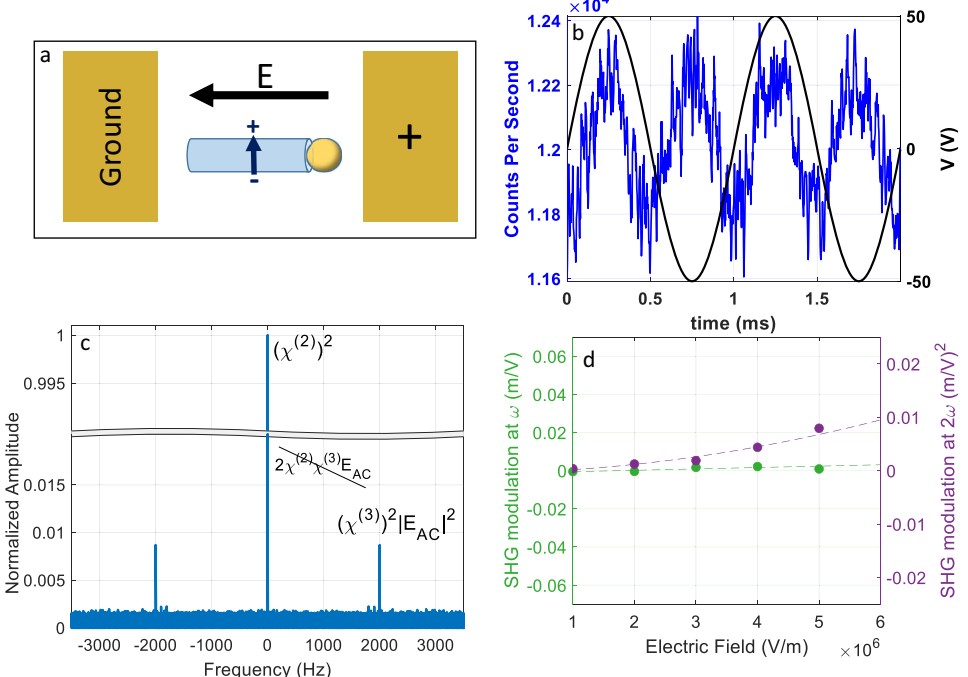

**Fig. 4 EFM-SHG from a nonpolar nanowire. a** A scheme of a nonpolar nanowire between two electrodes. The *c*-axis of the ZnO crystal (blue) is perpendicular to the nanowire's growth axis and to the external electric field (black). **b** Applied AC voltage (black) and the time-averaged optical SHG signal (blue) from a nonpolar nanowire showing oscillations at a double frequency. **c** FFT spectrum of the EFM-SHG. The frequency of the oscillating electric field is 1 kHz. Optical response is observed at 2 kHz. **d** Amplitude of the signal at 1 kHz (within the noise level) (green) and 2 kHz (purple) from the FFT spectrum as a function of the calculated electric field magnitude. SHG modulation at $\omega$ and $2\omega$ is presented in the same scale as in Fig. 2.

frequency. From Eq. (4) we indeed expect the optical signal at the SH frequency under an oscillating external field to contain three components (Fig. 3a): The first term at 0 Hz is proportional to $[\chi^{(2)}]^2$ and for a non-centrosymmetric material at appropriate polarizations it is the dominant part, which is not affected by the applied field. In our measurement configuration, both excitation and collection polarizations are along the *c*-axis, therefore probing $\chi^{(2)}_{zzz}$ (or $d_{33}$). The second term is at the applied AC frequency $\omega = 1$ kHz, and arises from the term $2\chi^{(2)}\chi^{(3)}E_{AC}$. The third term, in Eq. (4), $[\chi^{(3)}E_{AC}]^2$, becomes measurable under sufficiently strong fields, and appears as a peak at $2\omega$ in the FFT spectrum (Fig. 3a). Note that all three terms depend on the laser power as $|E^2(\omega)|^2$. The annotation in Fig. 3a is taken directly from Eq. (4) for clarity, the actual Fourier amplitudes differ slightly (see SI).

In Fig. 3b, c, we present the FFT components at $\omega$ (green) and $2\omega$ (purple) for right- and left-growing nanowires, respectively, as a function of the applied electric field. As predicted by Eq. (4), the FFT second term ($\omega$) and third term ($2\omega$) exhibit linear and quadratic dependence on the applied field, respectively. The electric field intensity $E_{AC}$ is estimated by dividing the amplitude of the applied AC voltage ($V_{AC}$) by the distance between the electrodes (10 μm). Each point represents 5 min of integration to allow detection of signals at low fields. For both the $\omega$ (green) and $2\omega$ (purple) components the corresponding term is normalized by the DC amplitude (0 Hz) in order to exclude artefacts due to laser power fluctuations over the entire length of the measurement set. For the linear $\omega$ plots (green), negative amplitude values indicate that the FFT component is anti-phase with respect to the applied field. This gives rise to a positive or negative slope depending on the internal dipole orientation of the nanowire in correlation with the in- and anti-phase behaviour discussed above. The amplitude of the $2\omega$ term (purple) shows a clear quadratic dependence on the electric field, as expected from Eq. (4). The application of an

oscillating AC field allows for easy separation of the different components similarly as in a lock-in setup. The same experiment under a constant field (DC) would require long acquisitions to overcome shot-noise, fluctuations in laser power and sample drifts.

To further test the validity of this method, we use it to probe SHG modulation along different, nonpolar, crystal axes. To achieve such configuration, nonpolar nanowires were grown on M-plane sapphire and gold electrodes were patterned as described above (Fig. 4a). In this case, the *c*-axis lies in-plane but perpendicular to the growth axis of the nanowire and to the electric field, and the matrix element of the SHG response along the nanowire is essentially zero[31]. This configuration is very similar to EFISH in a centrosymmetric material, where an electric field induces the otherwise zero SHG. In this case, we do not expect an optical response at $\omega$ but only at $2\omega$. Figure 4b shows the SH optical signal (blue) (averaged in time, see 'Methods') with respect to the supplied AC voltage (black). The optical signal in this case oscillates at double the frequency of the applied AC field, unlike in the case of polar nanowires (Fig. 3). Figure 4c shows the expected $2\omega$ component in the FFT spectrum. In Fig. 4d, we present the FFT components at $\omega$ (green) and $2\omega$ (purple), as a function of the applied electric field. The analysis is done in a similar manner as for the polar nanowires. As expected for a nonpolar nanowire, the amplitude at $\omega$ is below the noise level and the FFT amplitude at $2\omega$ exhibits a quadratic dependence on the applied field. As is apparent by the magnitude of the amplitude of the 0 Hz component in Fig. 4c, significant SH intensity is present even when no electric field is applied, this is most likely due to impurity of the excitation polarization. The excitation laser contains a polarization component perpendicular to the NW's growth axis and, since this is the most responsive axis of the crystal, SHG occurs and is detected in the experiment.

By performing field-strength-dependent measurements, we are able to extract the ratio between the nonlinear coefficients $\chi^{(3)}_{zzzz}$ and $\chi^{(2)}_{zzz}$ of these nanowires. The ratio is given by fits to a model derived from Eq. (4) (see SI for details of the model). For these nanowires we estimate: $\frac{\chi^{(3)}_{zzzz}(2\omega;\omega,\omega,0)}{\chi^{(2)}_{zzz}(2\omega;\omega,\omega)} = 35 \times 10^{-9}\,\text{m/V}$ (Table S2, Fig. S5). We could not find in the literature a reliable source for the value of $\chi^{(3)}_{zzzz}(2\omega;\omega,\omega,0)$. From known values[50] for $\chi^{(2)}_{zzz}(2\omega;\omega,\omega) = 15\,\text{pm/V}$ and our measured ratio we can estimate that $\chi^{(3)}_{zzzz}(2\omega;\omega,\omega,0) = 5.2\times10^4\,\text{pm}^2/\text{V}^2$ for ZnO WZ nanowires, this is in agreement with results from third harmonic generation experiments on ZnO thin films[56] $\chi^{(3)}_{zzzz}(3\omega;\omega,\omega,\omega) = 5.3\times10^4\,\text{pm}^2/\text{V}^2$. Here we only probe the $z$ coordinate of the WZ structure and perform a quantitative analysis, patterning a small electrode array around the NWs would give control over the spatial shape of the electric field enabling access to all crystal axes. Moreover, a THz source could, in principle, replace the applied electric field allowing for an all-optical measurement.

## Discussion

To conclude, we report the observation of electric-field-dependent interference in harmonic generation from polar materials and its use as a method for absolute polarity assignment at the nanoscale. This phenomenon was used to determine the absolute sign of polarity of surface-guided ZnO nanowires on sapphire. We found that the nanowires grow as oxygen-polar, regardless of their growth direction with respect to the underlying substrate, similarly to VLS vertical ZnO nanowires. Different epitaxial relations with the substrate turn out to not affect the absolute polarity of the surface-guided nanowires, but, on the other hand, lead to different morphology and quality of their alignment. Only a few seconds are required to extract the phase between the SH optical signal and the applied AC voltage, and thus determine the absolute sign of polarity. This method may not be limited to surface-guided nanowires, but could in principle be used for fast, simple and non-invasive assignment and imaging of absolute polarity of materials at the nanoscale, with the advantage that an interferometric setup is not needed, which makes the method significantly more robust. This could have many potential applications in materials and nanotechnology, including the characterization of semiconductors, whose polarity is an important parameter affecting their optoelectronic properties, as well as polycrystalline dielectric materials or ferroelectric materials, where the polarity distribution of the domains may be imaged directly.

## Methods

#### Guided growth of ZnO nanowires on R-plane sapphire

*Substrate preparation.* Wafers of R-plane ($1\bar{1}02$) and M-plane ($1\bar{1}00$) sapphire were purchased from Roditi International. A solution of Poly-L-lysine (0.1% w/v in $H_2O$, Sigma-Aldrich) is diluted in distilled water at a 1:10 ratio, dispersed on the substrate for 2 min, rinsed with water and dried with $N_2$. Next, a solution of gold nanoparticles is made by mixing (0.5% 50 nm + 0.5% 20 nm, Sigma-Aldrich) in distilled water. The solution is then dispersed on the substrate for 2 min, rinsed with water and dried with $N_2$. The substrate is then heated in a quartz tube at ambient atmosphere and 550 °C for 7 min to dispose of humidity and organic residues.

*Nanowire synthesis.* Nanowire growth was carried out in a CVD process in a quartz tube inside a tube furnace (Fig. S1). The Zn and O atoms were supplied from ZnO powder (Alpha Aesar 99.999%) mixed with graphite powder (Sigma-Aldrich 99.99%) at a 1:1 mass ratio, held at 1050 °C, while the samples were held 16 cm downstream at a temperature of 850 °C. $N_2$ was used as a carrier gas (99.999%, Gordon Gas, further filtered to reduce $O_2$ and $H_2O$ levels) at a flow of 500 sccm and a pressure of 400 mbar.

#### Electrodes fabrication

10 μm gap electrodes were fabricated by a standard photolithography process (MA/BA6 Karl Suss contact mask aligner). Alignment was done according to the flat of the wafer to form a configuration in which the nanowires are parallel to the applied electric field. Electron-beam evaporation (PVD, Telemark) was used to deposit Ti/Au (5 nm/100 nm), followed by liftoff in acetone.

#### Second-harmonic generation mapping and polarimetry

100 fs pulses @ 1000 nm at an 80 MHz repetition rate from a Ti:Sapphire laser (Coherent, Chameleon Ultra II) were passed through a half-wave plate and coupled into an inverted microscope (Zeiss, Axiovert 200 inverted microscope) and focused using an objective (Zeiss, Plan-Apochromat 10x/0.45NA). The epi-detected signal was filtered using a dichroic mirror (Thorlabs, DMSP950R), a band pass filter (Semrock, FF01-500/24-25), and a colour glass (Thorlabs, FGS900-KG3). A polarizer (analyser) was placed right before a multimode fibre (Thorlabs, FG050LGA). The SHG signal was detected by a single-photon avalanche photodiode (ID Quantique, ID100) which was connected to a time-correlated single-photon counting (TCSPC) system (Picoquant HydraHarp 400). The laser trigger output was connected to the TSCPS for synchronization.

Samples were placed on top of a cover slip such that the side on which the nanowires were grown was facing away from the microscope objective ("face up"). Under these conditions, the laser excitation was passed through the sapphire substrate and illuminated the nanowires "from bellow". In all experiments, the laser power was ~100 mW. For simplicity, samples were rotated in plane such that the nanowires growth axis was parallel to the $x$-axis in the lab frame up to a few degrees. Mapping was done by scanning the sample position using a piezo stage (Mad City Labs, Nano-BioS100) and measuring the SHG intensity while the excitation and collection polarizations were kept fixed. Polarimetry was performed by measuring the SHG intensity from a single point along the nanowire as a function of half-wave plate angle (changing the excitation polarization) for two analyser positions ($x$ and $y$ polarizations in the lab frame). In both mapping and polarimetry measurements the TSCPS was operated in histogram mode.

#### Electric-field-modulated second-harmonic generation

The sample, patterned with microelectrodes was placed face up on a cover slip to allow for easy needle probes access. The SH signal was collected from a diffraction-limited spot approximately in the middle of the nanowire and away from the electrodes. Probe positioners (Cascade Microtech, DPP-105-M-Al-S and Scientifica, Patch Star) were used to create electrical contact with any chosen electrode pair. A 1 kHz sine wave of various voltage amplitudes was supplied using an analogue voltage output device installed in a computer (National Instruments, PCI-6733). A synchronized "marker" was directed to the TCSPC to mark the beginning of each voltage cycle. The sine wave and markers were monitored with an oscilloscope. A time stamp was given to each photon and maker by the TCSPC.

#### Phase extraction for polarity assignment

The photon time trace was divided into segments of 5–30 s where each segment was synchronized by subtracting the time stamp of the first "marker" in the segment. Each segment was then binned, such that there were 100 "time bins" between every 2 markers, resulting in bin size of ~10 μs when a 1 kHz sine wave was applied. The discreet Fourier transform (DFT) of each segment was calculated and the phases of the 1 kHz ($\phi_{1\omega}$) and 2 kHz ($\phi_{2\omega}$) components were recorded directly. In addition, the absolute value of these components and of the DC (0 Hz) component were extracted for further analysis and are termed here as "Fourier amplitudes".

For complete details of the different methods refer to the Supplementary Information.

## Data availability

All datasets generated during and/or analysed during the current study are available from the corresponding author on reasonable request.

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

## Acknowledgements

The authors thank Dr. David Ehre and Prof. Igor Lubomirsky for helpful discussions. This research was supported by the European Research Council (ERC) Advanced Grants (No. 338849 and No. 669941), Israel Science Foundation (No. 2444/19) Helen and Martin Kimmel Centre for Nanoscale Science, Moskowitz Centre for Nano and Bio-Nano Imaging, the Crown Centre of Photonics and the Carolito Stiftung. E.J. holds the Drake Family Professorial Chair of Nanotechnology. D.O. is the incumbent of the Harry Weinrebe professorial chair of laser physics. R.B.Z. acknowledges funding from the Clore Foundation.

## Author contributions

All authors (E.J., R.B.Z., D.O. and O.B.E.) planned the research project and wrote the manuscript. R.B.Z. prepared the samples. O.B.E. and R.B.Z. conducted the optical measurements and data analysis. All authors (E.J., R.B.Z., D.O. and O.B.E) discussed the data and contributed to the manuscript.

## Competing interests

The authors declare no competing interests.
