## [Peer Review File · Nature Communications]

Reviewers' Comments:

Reviewer #1:

Remarks to the Author:

To be found in pdf format in section 4 (review attachments).

Mathematical expressions are more cumbersome using the present box.

This manuscript reports on an interesting and original, if not unique method to account for the sign of a dominant nonlinear quadratic tensor for a one-dimensional polar material. This approach is validated in the specific case of crystalline ZnO fibers grown over a sapphire surface in different conditions meant to vary the fiber polarity.

The manuscript features essentially two components: i) the optical set-up and the accompanying nonlinear optics considerations and ii) the interpretation of results from growth and crystallographic considerations.

The second part pertains to material sciences and appears to be a sound interpretation and discussion of the results from the nonlinear optical measurements which make for the central part of the manuscript.

The nonlinear optics part raises a fundamental question regarding the author's claim of a general method. It may be true that in the specific case of ZnO oriented fibers that the $\chi_{zzz}^{(2)}$ tensor component has the same sign as the dipole moment of rather dipole moment density μ_z but this must not be the case in general. Expressions of the quadratic tensor are built-up from transition dipole moments which amount to a sum of cubic products of transition dipole moments (weighed by frequency dispersion terms). Ground and excited state dipoles are part of such expansions (dipoles being special cases of transition dipoles $\mu_{e \rightarrow f}$ where $e = f$, should crystal symmetry dependent selection rules allow. Combinations of permanent dipoles of both fundamental and excited states may appear (for example their difference in a two-level frame), but not, except perhaps in very special cases unknown to this referee, to a permanent ground state dipole density (or an odd order product, here a cube, that could inform on its sign).

Indeed the case of a one-dimensional material where electron displacements are constrained to a single z axis and limiting to vertical transitions from valence to conduction bands, $\chi_{zzz}^{(2)}$ is then made of terms of the type $(\mu_{z;v \rightarrow v})^3$, $(\mu_{z;c \rightarrow c})^3$ and $\mu_{z;c \rightarrow c} |\mu_{z;v \rightarrow c}|^2$ where $\mu_{z;v \rightarrow c}$ stands for the transition dipole from the valence to the conduction band (along z and in a simplified two bands picture), $\mu_{z;v \rightarrow v}$ for the permanent dipole density along z of the valence state. Moreover, such cubic products of transition dipole moments are weighed by individual frequency dispersion terms which add to the sign determination and makes it a complex endeavor even in this simple case,

unless clear justification is provided. Indeed, continuing to refer to a simple valence to conduction band transitions, $\chi_{zzzz}^{(3)} = F(\omega_{opt}, \omega_{ac})(\mu_{v \rightarrow c})^4$ plus other similar terms, where $F(\omega_{opt}, \omega_{ac})$ stands for a dispersion term which may be negative or positive depending on the relative position of the optical field with respect to the valence to conduction band energy differences.

To summarize, it is not at all obvious why the sign of the (permanent) ground state dipole moment density should be the same as that of the measured $\chi_{zzz}^{(2)}\chi_{zzzz}^{(3)}$ (leaving aside the sign of the $\chi_{zzzz}^{(3)}$ which can be taken as positive, following a simple derivation along the same line of a two bands model as proposed before (this being left to the authors)).

Other than that, the optical set-up is clever and it is not altogether unreasonable to assign it the new label EFM-SHG so as to distinguish it from the ancient and well documented EFISH technique, although one could claim it actually belongs to this family in the broader sense. But in any case, this is a matter of taste and future can better tell, more than the other's somewhat biased decision, if this new label is justified and does take off.

The originality here consists in the beating of an already existing quadratic tensor $\chi_{zzz}^{(2)}$ with the cubic tensor contracted with the applied field $\chi_{zzzz}^{(3)}E$. This beating results from the intensity detector squaring of the induced polarization from both origins. More generally, this leads to three terms respectively at zero, once and twice the frequency of the applied field. Following the implicit rational that $\chi_{zzzz}^{(3)}$ is positive (although this does not appear to be clearly stated in the text, see above considerations), one can then infer an "absolute" sign for $\chi_{zzz}^{(2)}$ (which, there again, is not necessarily that of the dipole density, subject to further comments by the authors).

These grossly oversimplified considerations are just meant here to point-out a difficulty that the authors seem to have ignored and should clarify (subject to their choice of a possibly more appropriate model to account for the linear and nonlinear properties of ZnO).

The use by the authors of the FFT algorithm to extract in a quasi-instantaneous manner the three frequency components and their amplitudes is working in an impressive way and produce neat results, as shown for example in their highly convincing Fig. 3.

At a general level, the authors fail to mention in their introductory review among possibly alternative methods the electro-optic Pockels effect, which may entail the benefit of being lower order than a $\chi^{(2)}\chi^{(3)}$ effect, while providing similar results in an interferometric geometry as exemplified for example in Trinh et al., *Electro-optical interferometric microscopy of periodic and aperiodic ferroelectric structures*, Laser Photonics Rev. 9 (2), 214–223 (2015).

It might be also worthwhile to cite for SHG methods L. Le Xuan et al. *Balanced homodyne detection of second-harmonic generation from isolated subwavelength emitters*, Appl.Phys.Lett. 89, 121118, 2006

This being said, one cannot but praise the excellent nonlinear optic experimental part in this manuscript.

The authors might reinforce their case by performing modifications and additions as requested here and then go through a second row of revision and also, enlarge the perspective and relevance of their paper to make sure that it is found of interest by a broader readership (again, the validity of the method is established by exemplifying the case of ZnO, but further arguments, say theoretical, should be provided to support its full scope beyond this simple case)

Reviewer #2:

Remarks to the Author:

Summary:

The authors demonstrate a practical means for identifying the absolute polar order of ZnO nanowires using optical rectification, in which DC fields couple into second order nonlinear optical interactions. The approach, dubbed herein as electric-field modulated second harmonic generation (EFM-SHG) in this application space, addresses ambiguities in ZnO nanostructure design, as catalytic properties of such materials are sensitive to the exposed crystallographic interface. The authors demonstrate nicely the ability to modulate the SHG using an external field applied across individual nanowires and convincingly use FT analysis to aid in disentangling native [$\chi(2)$] and field-induced perturbations [$\chi(3)$] to the SHG. Although effects related to optical rectification such as those demonstrated in this work have been well established both theoretically and experimentally for more than 60 years, this work nicely demonstrates the utility of that framework for probing nanostructured materials. Overall, this manuscript is worthy of consideration for publication in Nature Communication, provided the authors make changes to better clarify the breadth of applications for the method and the novel NLO aspects of nanoscale materials relative to the wealth of prior work with analogous bulk NLO materials.

Comments:

1. Light-induced modulation of DC fields in second order NLO crystals (and the converse) was first demonstrated for macroscopic crystals by Ward and coworkers in 1965 (Phys. Rev. Lett.) using an approach virtually identical to that demonstrated by the authors in this study. The manuscript could be improved by additional content to clarify the specific differences arising from the reduction in scale relative to that previous work. Do new phenomena arise at nanoscale that are inaccessible from macroscopic analyses? A more detailed comparison and contrast is recommended.
2. The application demonstrated in this work relied on the presence of polar order along the long axis of the ZnO nanowire connected to electrodes. However, it is not clear whether broad classes of polar wurzite structures are likely to produce comparable sensitivity for other ionic crystals. Furthermore, nanowires comprised of centrosymmetric lattices will not produce SHG or exhibit polar order. Additional information should be added to the manuscript to better clarify the application space in which this method may be used in practice (e.g., when electric contacts may be inaccessible).
3. The manuscript could benefit from restructuring of the content to improve flow. Given the short nature of a communication, the authors should group complete discussions of each topic together to improve readability (e.g., discriminating $\chi(2)$ and $\chi(3)$ contributions, quantifying the role of the substrate, and ultimately determining polar order).

Reviewer #3:

Remarks to the Author:

In their paper Ben-Zvi et al. present the polarity-dependent nonlinear optics of nanowires under an electric field. Material characteristics of polar materials as ZnO are determined by their polar orientation. For the planar growth of ZnO NWs by means of VLS method, it is difficult and time-consuming to determine the crystal polarity with standard methods. Often these methods are invasive and preventing a further processing of the measured NWs. Ben-Zvi et al. are introducing a new non-invasive methodology electric-field-modulated second-harmonic generation (EFM-SHG) for the determination of the polar orientation of ZnO NWs on r-plane sapphire substrates. This technique provides fundamental information whether the polarity is defined by the atomic arrangement of the substrate or by the growth direction. While I generally recommend this manuscript, the following remarks will help to improve the paper before publication.

1

Can the authors explain the origin of the anion polar orientation for VLS grown NWs more in details (cf. page 6 and page 13 in manuscript)? Is there an explanation in literature, why the polarity is depending on the growth mechanism (self-catalyzed vs. VLS)?

#2

In Figure 1 it is clearly visible that NWs, which are growing to the right, are kinked. I understand that this originates from the asymmetry of the sapphire surface. Nevertheless, this would underline a surface dependence of the growth direction, because kinking means change of growth direction/growth front. How does the kinking agree with the polarity and the presented model? Is this due to a competition between surface-determined and growth direction-determined polarity?

#3

Right-growing NWs are not linear, but kinked. This means a change of the growth direction of the NW. Please, could the authors explain, how that is included in their method and how does the regular variation of the growth direction affect the EFM-SHG signal?

#4

To me, it would be clearer, if the authors could change the coloring of the red and the blue arrow in Figure 2. Then it would correspond to the color scheme in Figure 1 and Figure 3.

#5

On page 12, the authors are writing that the precise spotting of the Au catalyst is difficult. Why was not energy-dispersive X-ray spectroscopy (EDS) used for locating the Au catalyst?

#6

Nicely, the authors show the measurements of 18 NWs and are explaining their used method in details. Nevertheless, I would favor if the authors can at least verify their outcomes by the use of one standard procedure for one to two NWs.

Response to reviewers

We would first like to thank all reviewers for their overall positive view of the manuscript, for the constructive comments and for their help in improving the manuscript. The main concerns by the reviewers had to do with the oversimplified explanation of the relation between crystal polarity and the relative sign of the second and third order nonlinear terms (Reviewer #1), with better placement of the work in the context of the Pockels effect (Reviewers #1 and #2) and with the interpretation of the nanowire direction (Reviewer #3). In the resubmission, we have addressed all these. We humbly agree with the comments of Reviewer #1 regarding the generality of the relation between the crystal dipole and the relative phase, although we note that this does not change the results on ZnO or the conclusions of the work. It simply calls for the use of tabulated literature $\chi^{(2)}$ and $\chi^{(3)}$ in the interpretation. We have revised the text in the introduction to highlight the differences between EFM-SHG and Pockels microscopy. Finally, we clarified the explanation on the effect of growth direction on nanowire morphology. We hope that with these changes, you find the manuscript suitable for publication in *Nature Communications*.

Below is our point-by-point response to the reviewers. We have marked in red the issues raised by the reviewers, and in blue our response to them.

Reviewer #1 (Remarks to the Author):

This manuscript reports on an interesting and original, if not unique method to account for the sign of a dominant nonlinear quadratic tensor for a one-dimensional polar material. This approach is validated in the specific case of crystalline ZnO fibers grown over a sapphire surface in different conditions meant to vary the fiber polarity.

The manuscript features essentially two components: i) the optical set-up and the accompanying nonlinear optics considerations and ii) the interpretation of results from growth and crystallographic considerations.

The second part pertains to material sciences and appears to be a sound interpretation and discussion of the results from the nonlinear optical measurements which make for the central part of the manuscript.

“The nonlinear optics part raises a fundamental question regarding the author’s claim of a general method. It may be true that in the specific case of ZnO oriented fibers that the $\chi_{zzz}^{(2)}$ tensor component has the same sign as the dipole moment of rather dipole moment density μ but this must not be the case in general.”

Expressions of the quadratic tensor are built-up from transition dipole moments which amount to a sum of cubic products of transition dipole moments (weighed by frequency dispersion terms). Ground and excited state dipoles are part of such expansions (dipoles being special cases of transition dipoles $\mu_{e \rightarrow f}$ where $e = f$, should crystal symmetry dependent selection rules allow. Combinations of permanent dipoles of both fundamental and excited states may appear (for example their difference in a two-level frame), but not, except perhaps in very special cases unknown to this referee, to a permanent ground state dipole density (or an odd order product, here a cube, that could inform on its sign).

Indeed the case of a one-dimensional material where electron displacements are constrained to a single z axis and limiting to vertical transitions from valence to conduction bands, $\chi^{(2)}_{zzz}$ is then made of terms of the type $(\mu_{z;v \rightarrow v})^3$, $(\mu_{z;c \rightarrow c})^3$ and $(\mu_{z;c \rightarrow c})(\mu_{z;v \rightarrow c})^2$ where $\mu_{z;v \rightarrow c}$ stands for the transition dipole from the valence to the conduction band (along z and in a simplified two bands picture), $\mu_{z;v \rightarrow v}$ for the permanent dipole density along z of the valence state. Moreover, such cubic products of transition dipole moments are weighed by individual frequency dispersion terms which add to the sign determination and makes it a complex endeavor even in this simple case, unless clear justification is provided. Indeed, continuing to refer to a simple valence to conduction band transitions, $\chi^{(3)}_{zzzz} = F(\omega_{\text{opt}}, \omega_{\text{ac}})(\mu_{v \rightarrow c})^4$ plus other similar terms, where $F(\omega_{\text{opt}}, \omega_{\text{ac}})$ stands for a dispersion term which may be negative or positive depending on the relative position of the optical field with respect to the valence to conduction band energy differences.

To summarize, it is not at all obvious why the sign of the (permanent) ground state dipole moment density should be the same as that of the measured $\chi^{(2)}_{zzz}$ $\chi^{(3)}_{zzzz}$ (leaving aside the sign of the $\chi^{(3)}_{zzzz}$ which can be taken as positive, following a simple derivation along the same line of a two bands model as proposed before (this being left to the authors).

We humbly agree with the reviewer on this point. In general, we note that (as discussed by the reviewer) under nonresonant conditions the third order nonlinear contribution is polarized along the direction of the DC electric field. However, as stated by the reviewer, the sign of the second order nonlinear term need not generally align with the internal dipole. We have modified the discussion on this point, and added references showing that for the particular case of ZnO our description was correct. We note that the sign of the second order term is tabulated for numerous materials and thus this point does not significantly reduce the generality or applicability of our scheme.

Other than that, the optical set-up is clever and it is not altogether unreasonable to assign it the new label EFM-SHG so as to distinguish it from the ancient and well documented EFISH technique, although one could claim it actually belongs to this family in the broader sense. But in any case, this is a matter of taste and future can better tell, more than the other's somewhat biased decision, if this new label is justified and does take off.

The originality here consists in the beating of an already existing quadratic tensor $\chi^{(2)}_{zzz}$ with the cubic tensor contracted with the applied field $\chi^{(3)}_{zzzz} E$. This beating results from the intensity detector squaring of the induced polarization from both origins. More generally, this leads to three terms respectively at zero, once and twice the frequency of the applied field. Following the implicit rationale that $\chi^{(3)}_{zzzz}$ is positive (although this does not appear to be clearly stated in the text, see above considerations), one can then infer an “absolute” sign for $\chi^{(2)}_{zzz}$ (which, there again, is not necessarily that of the dipole density, subject to further comments by the authors).

These grossly oversimplified considerations are just meant here to point-out a difficulty that the authors seem to have ignored and should clarify (subject to their choice of a possibly more appropriate model to account for the linear and nonlinear properties of ZnO).

We appreciate the positive comments by the referee on the novelty of the scheme we present. We completely agree that it belongs in the “EFISH family”, but felt the need to use another term as EFISH has practically become synonymous with measuring SHG from centrosymmetric or amorphous materials. As discussed above, and as noted by the referee, we now explicitly mention the assumption regarding the sign of $\chi^{(3)}_{zzzz}$ being positive and have corrected the discussion on the relation of the sign of the beat signal between the $\chi^{(3)}_{zzzz}$ and $\chi^{(2)}_{zzz}$ terms and the dipole orientation for the particular case of ZnO.

The use by the authors of the FFT algorithm to extract in a quasi-instantaneous manner the three frequency components and their amplitudes is working in an impressive way and produce neat results, as shown for example in their highly convincing Fig. 3.

At a general level, the authors fail to mention in their introductory review among possibly alternative methods the electro-optic Pockels effect, which may entail the benefit of being lower order than a $\chi^{(2)}\chi^{(3)}$ effect, while providing similar results in an interferometric geometry as exemplified for example in Trinh et al., *Electro-optical interferometric microscopy of periodic and aperiodic ferroelectric structures*, *Laser Photonics Rev.* 9 (2), 214–223 (2015).

We have added reference to Pockels imaging which is indeed another technique capable of determining the absolute polarity via an interference term (between the first and second order susceptibilities). We noted in the text the significant difference between the implementation of Trinh et al, where interferometric detection was still needed and the current setup which leads to an intensity modulation of the SHG, alleviating the need for interferometry.

It might be also worthwhile to cite for SHG methods L. Le Xuan et al. *Balanced homodyne detection of second-harmonic generation from isolated subwavelength emitters*, Appl.Phys.Lett. 89, 121118, 2006

Reference was added to Le Xuan et al.

This being said, one cannot but praise the excellent nonlinear optic experimental part in this manuscript.

The authors might reinforce their case by performing modifications and additions as requested here and then go through a second row of revision and also, enlarge the perspective and relevance of their paper to make sure that it is found of interest by a broader readership (again, the validity of the method is established by exemplifying the case of ZnO, but further arguments, say theoretical, should be provided to support its full scope beyond this simple case)

We appreciate this suggestion but feel that a broader theoretical discussion is beyond the scope of this predominantly experimental work.

Reviewer #2 (Remarks to the Author):

Summary:

The authors demonstrate a practical means for identifying the absolute polar order of ZnO nanowires using optical rectification, in which DC fields couple into second order nonlinear optical interactions. The approach, dubbed herein as electric-field modulated second harmonic generation (EFM-SHG) in this application space, addresses ambiguities in ZnO nanostructure design, as catalytic properties of such materials are sensitive to the exposed crystallographic interface. The authors demonstrate nicely the ability to modulate the SHG using an external field applied across individual nanowires and convincingly use FT analysis to aid in disentangling native $[\chi(2)]$ and field-induced perturbations $[\chi(3)]$ to the SHG. Although effects related to optical rectification such as those demonstrated in this work have been well established both theoretically and experimentally for more than 60 years, this work nicely demonstrates the utility of that framework for probing nanostructured materials. Overall, this manuscript is worthy of consideration for publication in Nature Communication, provided the authors make changes to better clarify the breadth of applications for the method and the novel NLO aspects of nanoscale materials relative to the wealth of prior work with analogous bulk NLO materials.

Comments:

1. Light-induced modulation of DC fields in second order NLO crystals (and the converse) was first demonstrated for macroscopic crystals by Ward and coworkers in 1965 (Phys. Rev. Lett.) using an approach virtually identical to that demonstrated by the authors in this study. The manuscript could be improved by additional content to clarify the specific differences arising from the reduction in scale relative

to that previous work. Do new phenomena arise at nanoscale that are inaccessible from macroscopic analyses? A more detailed comparison and contrast is recommended.

We are not sure which of the series of papers of Ward and coworkers on optical rectification and related phenomena the reviewer relates to. In any case, the approach here is different since it involves interference between a second and a third order nonlinear process (rather than, for example, an interference between the dielectric tensor (first order) and second order. Thus, it is possible to obtain an amplitude modulation of the SHG signal and read out the signal without interferometric measurements (see also response to Reviewer #1 above). There is no significant difference between nanostructures and more macroscopic objects here, except for the fact that low voltages are sufficient to get a sizable response.

2. The application demonstrated in this work relied on the presence of polar order along the long axis of the ZnO nanowire connected to electrodes. However, it is not clear whether broad classes of polar wurzite structures are likely to produce comparable sensitivity for other ionic crystals. Furthermore, nanowires comprised of centrosymmetric lattices will not produce SHG or exhibit polar order. Additional information should be added to the manuscript to better clarify the application space in which this method may be used in practice (e.g., when electric contacts may be inaccessible).

The second order nonlinear susceptibility of ZnO is indeed relatively high (although there are many materials with higher $\chi(2)$ coefficients). Notably, however, the measurements presented here present a very high SNR at moderate integration times, and the ratio of $\chi(2)$ to $\chi(3)$ would likely be more favorable for materials with a lower second order susceptibility. As such, neither the magnitude of the needed field nor the signal strength seem to be significantly limiting factors. As for centrosymmetric lattices – we are not sure what the reviewer refers to here. Centrosymmetric materials are not polar and as such there is no issue of determining their polarity.

3. The manuscript could benefit from restructuring of the content to improve flow. Given the short nature of a communication, the authors should group complete discussions of each topic together to improve readability (e.g., discriminating $\chi(2)$ and $\chi(3)$ contributions, quantifying the role of the substrate, and ultimately determining polar order).

This is a good suggestion. We have grouped discussions of each topic together under different sub-headings. We believe that this has improved the reading flow of the manuscript.

Reviewer #3 (Remarks to the Author):

In their paper Ben-Zvi et al. present the polarity-dependent nonlinear optics of nanowires under an electric field. Material characteristics of polar materials as ZnO are determined by their polar orientation. For the planar growth of ZnO NWs by means of VLS method, it is difficult and time-consuming to determine the crystal polarity with standard methods. Often these methods are invasive and preventing a further processing of the measured NWs. Ben-Zvi et al. are introducing a new non-

invasive methodology electric-field-modulated second-harmonic generation (EFM-SHG) for the determination of the polar orientation of ZnO NWs on r-plane sapphire substrates. This technique provides fundamental information whether the polarity is defined by the atomic arrangement of the substrate or by the growth direction. While I generally recommend this manuscript, the following remarks will help to improve the paper before publication.

1

Can the authors explain the origin of the anion polar orientation for VLS grown NWs more in details (cf. page 6 and page 13 in manuscript)? Is there an explanation in literature, why the polarity is depending on the growth mechanism (self-catalyzed vs. VLS)?

Yes, the mechanisms that lead to the anion polar orientation for VLS grown NWs as opposed to the cation-polar orientation for catalyst-free growth of NWs, have been thoroughly reviewed and explained in a recent review: de la Mata, M. et al. The Role of Polarity in Nonplanar Semiconductor Nanostructures. *Nano Lett.* **19**, 3396–3408 (2019), which was cited in the introduction. In the revised manuscript, we have added a sentence in the discussion that refers again to this review more specifically when explaining the important role of the catalyst droplet in determining nanowire polarity (page 13 in the revised manuscript).

#2

In Figure 1 it is clearly visible that NWs, which are growing to the right, are kinked. I understand that this originates from the asymmetry of the sapphire surface. Nevertheless, this would underline a surface dependence of the growth direction, because kinking means change of growth direction/growth front. How does the kinking agree with the polarity and the presented model? Is this due to a competition between surface-determined and growth direction-determined polarity?

These are all very interesting questions, and we completely agree with the reviewer's interpretation that kinking may be a sign of competition between surface-determined and growth direction-determined polarity. Moreover, the fact that the latter wins here for ZnO NWs on R-plane sapphire, does not mean there could not be a different outcome for other substrate-material combinations. We have added a few lines of discussion about this to the relevant paragraph (page 13 in the in the revised manuscript). Regarding the model, we should stress that there is yet no theoretical model in literature that explains or predicts the direction of epitaxial surface-guided nanowires. This is another reason why this report is important, because it provides more detailed experimental data that could be used to develop and test such model.

#3

Right-growing NWs are not linear, but kinked. This means a change of the growth direction of the NW. Please, could the authors explain, how that is included in their

method and how does the regular variation of the growth direction affect the EFM-SHG signal?

We agree with the reviewer that the kinked morphology could in principle affect the optical signal. However, since the kinked segments are relatively much shorter than the ones along the main direction, we opted to neglect this possible effect, which could greatly complicate the analysis. A sentence explaining this has been added to the same paragraph where we discuss the kinked morphology as in relation to the previous comment.

#4

To me, it would be clearer, if the authors could change the coloring of the red and the blue arrow in Figure 2. Then it would correspond to the color scheme in Figure 1 and Figure 3.

The reviewer is completely right and we thank for noticing this lack of consistency in the coloring. However, following the convention in Physics that dipole moment arrows go from - to + (unlike the opposite convention in Chemistry from + to -), we have left Figure 2 as is, and changed the color scheme in Figures 1 and Figure 3 to be consistent with Figure 2. All the figures are now consistent with each other, as well as with the legends and the text.

#5

On page 12, the authors are writing that the precise spotting of the Au catalyst is difficult. Why was not energy-dispersive X-ray spectroscopy (EDS) used for locating the Au catalyst?

In most cases it is not difficult to spot the Au catalyst at the end of each nanowire, except for a few cases, especially when the growing end is covered by the Au electrode. In this case, EDS would certainly not help to spot the Au catalyst, as all the end segment of the nanowire is covered with Au. We have added a few words to clarify this situation, referring to the Supporting Information that explains it. Regarding the use of EDS to identify the Au catalyst, we have tried it in other samples, but EDS is not always sensitive enough to reliably detect Au from the nanoparticle at the end of each nanowire. In the present case, as explained before, the use of EDS had no need or advantage.

#6

Nicely, the authors show the measurements of 18 NWs and are explaining their used method in details. Nevertheless, I would favor if the authors can at least verify their outcomes by the use of one standard procedure for one to two NWs.

We have tried very hard for more than half a year to determine the polarity of the nanowires by the standard electron microscopy method (STEM-HAADF), but it was extremely difficult to cut a thin enough lamella along the relatively narrow ZnO NWs, as we could do in the past for thicker NWs (e.g. GaN NWs on sapphire, in

Science 2011, 333, 1003). We also tried to image the ZnO NWs from the top after thinning the sapphire substrate from the back by mechanical polishing, but all the samples were damaged in the process. All these difficulties underscore the importance of the optical method described in the manuscript, especially for nanocrystals that are strongly bound to a thick substrate that is opaque to TEM. The polarities that we determined by this method are strongly supported by the consistency between the 18 measurements, the quantitative consistency with the physical model, and the consistency with the anion-polar direction that is reported in literature for Au-catalyzed ZnO nanowires in particular, and nanowires of II-VI compound semiconductors in general.

Reviewers' Comments:

Reviewer #1:

Remarks to the Author:

The main issue raised in my initial report, namely the sign of the cubic susceptibility vis à vis dipole orientation has been answered in a satisfactory way.

The manuscript appears now fully consistent and appropriate for publication in one of the Nature journals. Nature Communication may not be the best choice in view of the mainly material science oriented objective of this work. The physics part is an interesting and clever variation of an otherwise generic NLO experimental scheme, thus belonging to the domain of instrumentation for material characterisation. The material science objective and the results therefrom are of greater relevance and concern mainly the semiconductor material science community.

Nature material might seem therefore a more appropriate media.

Reviewer #2:

Remarks to the Author:

The revisions to the manuscript address many of the questions raised during the prior review. The authors are encouraged to make additional minor changes to better place the novelty in their studies within the context of prior DC-field modulated nonlinear optics research. Following these straightforward changes, the editors are encouraged to publish the revised manuscript.

Regarding my prior reference to the 1965 Phys. Rev. Lett. article by Ward and coworkers, my apologies for the typo; it should have been 1962, in which a DC potential was measured across an SHG-active crystal upon laser exposure (essentially, the inverse of the present study): "Optical Rectification" Bass, Ward, Weinreich, and Franken, Phys. Rev. Lett., v9, p446, 1962. In follow-up work by Bloembergen and coworkers, the experimental demonstration of modulation of SHG by an applied electric field was expressed as the interference between a second order NLO response and a third-order DC-field-induced perturbation: "Nonlinear Electroreflectance in Silicon and Silver" Lee, Chang, Bloembergen, Phys. Rev. Lett, v18, p167, 1967 (Eq. 1 in the 1967 article mirrors Eq. 3 in the present manuscript). More recently (but still nearly 30 years ago), interference between $\chi^{(2)}$ and DC-field driven $\chi^{(3)}$ sources of SHG was first used to determine absolute polar orientation of water molecules at charged interfaces (Ong, Zhao, Eienthal, Chem. Phys. Lett., v191, p327, 1992). For a more definitive recent example of electric-field modulated SHG to determine polar order, the authors are referred to (Nature Comm. v7, article 13587, 2016) by Ohno, Saslow Wang, Geiger, and Eienthal. The measurements demonstrated by the authors differ from this prior body of work in two notable respects: i) application of the approach for the analysis of nanowires, and ii) use of sinusoidally modulated electric fields and harmonics analysis. These are sufficiently interesting and enabling advances to warrant publication in Nature Communications. However, the authors are still encouraged to rewrite the manuscript to acknowledge the foundational work by these early NLO pioneers and avoid descriptors such as "... a property to be added to the series" in the Abstract and "...we report the observation of an electro-optical property of polar materials and its use..." in the last paragraph. Such claims might lead readers to believe that electric field-modulated SHG was first experimentally observed in the present study, rather than >50 years ago in 1967 by Bloembergen, just a handful of years after the invention of the laser.

Reviewer #3:

Remarks to the Author:

The paper uses a unique method to get important information about the polarity, including their absolute sign, of surface guided nanowires. The approach is based on nonlinear electro-optical phenomenon of EFM-SHG under varied electric field intensities in combination with sound theory evaluations. The paper shows that the EFM-SHG can be used as a nondestructive tool for polarity

mapping of nanostructures.

The paper is significantly improved and my concerns are sufficiently addressed. I approve publication of the paper in its revised form. However, I am not completely convinced that Nature Communications is the correct journal. The focus of the paper is mainly on establishing a new and non-destructive method for determination polarity of surface guided grown nanowires including their absolute sign. In principle it represents a measurement method. I think the paper would fit much better into Nature Methods.

Polarity-dependent nonlinear optics of nanowires under electric field

Nature Communications manuscript NCOMMS-20-22488A

Point-by-point response to the reviewers' comments

Reviewer #1 (Remarks to the Author):

The main issue raised in my initial report, namely the sign of the cubic susceptibility vis à vis dipole orientation has been answered in a satisfactory way. The manuscript appears now fully consistent and appropriate for publication in one of the Nature journals. Nature Communication may not be the best choice in view of the mainly material science oriented objective of this work. The physics part is an interesting and clever variation of an otherwise generic NLO experimental scheme, thus belonging to the domain of instrumentation for material characterisation. The material science objective and the results therefrom are of greater relevance and concern mainly the semiconductor material science community. Nature material might seem therefore a more appropriate media.

We thank the reviewer his/her positive comments.

Reviewer #2 (Remarks to the Author):

The revisions to the manuscript address many of the questions raised during the prior review. The authors are encouraged to make additional minor changes to better place the novelty in their studies within the context of prior DC-field modulated nonlinear optics research. Following these straightforward changes, the editors are encouraged to publish the revised manuscript.

We thank the referee for his/her positive recommendation

Regarding my prior reference to the 1965 Phys. Rev. Lett. article by Ward and coworkers, my apologies for the typo; it should have been 1962, in which a DC potential was measured across an SHG-active crystal upon laser exposure (essentially, the inverse of the present study): "Optical Rectification" Bass, Ward, Weinreich, and Franken, Phys. Rev. Lett., v9, p446, 1962.

We thank the referee for bringing this up. However, optical rectification is a second order process whereas the effect we discuss here relates to a third order process. Hence we do not think this reference is appropriate.

In follow-up work by Bloembergen and coworkers, the experimental demonstration of modulation of SHG by an applied electric field was expressed as the interference between a second order NLO response and a third-order DC-field-induced perturbation: "Nonlinear Electroreflectance in Silicon and Silver" Lee, Chang, Bloembergen, Phys. Rev. Lett, v18, p167, 1967 (Eq. 1 in the 1967 article mirrors Eq. 3 in the present manuscript).

We thank the referee for pointing this out and added this reference in the introduction, highlighting why it also does not overlap with what we show in this manuscript (despite the seeming similarity in equations):

“We note that electric field modulation of SHG in centrosymmetric materials exhibiting SHG of an electric quadrupole or magnetic dipole origin does not lead to such an interference since in this case the static and EFISH components are at quadrature with each other for nonabsorbing media [Lee et al., PRL 18, 167, 1967].”

More recently (but still nearly 30 years ago), interference between $\chi(2)$ and DC-field driven $\chi(3)$ sources of SHG was first used to determine absolute polar orientation of water molecules at charged interfaces (Ong, Zhao, Eienthal, Chem. Phys. Lett., v191, p327, 1992). For a more definitive recent example of electric-field modulated SHG to determine polar order, the authors are referred to (Nature Comm. v7, article 13587, 2016) by Ohno, Saslow Wang, Geiger, and Eienthal.

These references are indeed more relevant, and we added them to the text around Equation 3. We note, however, that no external electric field is applied there (rather a pH change, chemically inducing a surface dipole) and that, unlike our work, it is not a bulk effect.

The measurements demonstrated by the authors differ from this prior body of work in two notable respects: i) application of the approach for the analysis of nanowires, and ii) use of sinusoidally modulated electric fields and harmonics analysis. These are sufficiently interesting and enabling advances to warrant publication in Nature Communications. However, the authors are still encouraged to rewrite the manuscript to acknowledge the foundational work by these early NLO pioneers and avoid descriptors such as “... a property to be added to the series” in the Abstract and “...we report the observation of an electro-optical property of polar materials and its use...” in the last paragraph. Such claims might lead readers to believe that electric field-modulated SHG was first experimentally observed in the present study, rather than >50 years ago in 1967 by Bloembergen, just a handful of years after the invention of the laser.

We have slightly modified the wording in the introduction and the discussion to better reflect the pioneering work of Bloembergen and others.

Reviewer #3 (Remarks to the Author):

The paper uses a unique method to get important information about the polarity, including their absolute sign, of surface-guided nanowires. The approach is based on the nonlinear electro-optical phenomenon of EFM-SHG under varied electric field intensities in combination with sound theory evaluations. The paper shows that the EFF-SHG can be used as a nondestructive tool for polarity mapping of nanostructures.

The paper is significantly improved and my concerns are sufficiently addressed. I approve publication of the paper in its revised form. However, I am not completely convinced that Nature Communications is the correct journal. The focus of the paper is mainly on establishing a new and non-destructive method for determination of polarity of surface-guided grown nanowires including their absolute sign. In principle it represents a measurement method. I think the paper would fit much better into Nature Methods.

We thank the reviewer for his/her positive comments.